# Influence of Bio-Based Plasticizers on the Properties of NBR Materials

**DOI:** 10.3390/ma13092095

**Published:** 2020-05-01

**Authors:** Md Mahbubur Rahman, Katja Oßwald, Katrin Reincke, Beate Langer

**Affiliations:** 1Department of Engineering and Natural Sciences, Hochschule Merseburg-University of Applied Sciences, Eberhard-Leibnitz-Straße 2, 06217 Merseburg, Germany; beate.langer@hs-merseburg.de; 2Polymer Service GmbH Merseburg, Associate institute of Hochschule Merseburg-University of Applied Sciences, 06217 Merseburg, Germany; katja.osswald@psm-merseburg.de (K.O.); katrin.reincke@psm-merseburg.de (K.R.)

**Keywords:** bio-oil, bio-based plasticizer, eco-friendly plasticizer, acrylonitrile-butadiene rubber, NBR, mechanical testing, thermo-oxidative aging

## Abstract

A high number of technical elastomer products contain plasticizers for tailoring material properties. Some additives used as plasticizers pose a health risk or have inadequate material properties. Therefore, research is going on in this field to find sustainable alternatives for conventional plasticizers. In this paper, two modified bio-based plasticizers (epoxidized esters of glycerol formal from soybean and canola oil) are of main interest. The study aimed to determine the influence of these sustainable plasticizers on the properties of acrylonitrile–butadiene rubber (NBR). For comparison, the influence of conventional plasticizers, e.g., treated distillate aromatic extract (TDAE) and Mesamoll^®^ were additionally investigated. Two types of NBR with different ratios of monomers formed the polymeric basis of the prepared elastomers. The variation of the monomer ratio results in different polarities, and therefore, compatibility between the NBR and plasticizers should be influenced. The mechanical characteristics were investigated. In parallel, dynamic mechanical analysis (DMA) and thermogravimetric analysis (TGA) were performed and filler macro-dispersion was determined. Bio-based plasticizers were shown to have better mechanical and thermal properties compared to conventional plasticizers. Further, thermo-oxidative aging was realized for 500 h, and afterwards, mechanical characterizations were done. It was observed that bio-based plasticizers have almost the same aging properties compared to conventional plasticizers.

## 1. Introduction

Acrylonitrile-butadiene rubber (NBR) is a synthetic unsaturated statistical copolymer of acrylonitrile (ACN) and butadiene. NBR has good oil and chemical resistance [1]. It is used in the automotive and petroleum industries for oil and engine fuel transport equipment, for machinery pumps and in disposable nonlatex gloves [2]. It is a perfect choice for sealing applications due to its resistive properties, e.g., as O-rings due to its oil resistance [3]. Excellent processability is also a concern. Despite application, the field varies with the proportion of ACN content and molecular weight of the polymer. It has excellent resistance to petroleum products over a wide temperature range [3]. NBR typically contains extender oil that works as a plasticizer. Unfilled NBR has a low level of mechanical properties; thus, filler is incorporated into the NBR. This increases the viscosity and reduces the internal plasticization, so the addition of an external plasticizer is needed [4]. Various types of oil can be added to NBR compounds to reduce viscosity, improve processing properties, increase low-temperature flexibility, and reduce production costs [5].

The International Union of Pure and Applied Chemistry (IUPAC) developed a universally accepted definition of plasticizers in 1951: “A plasticizer is a substance or material incorporated in a material to increase its flexibility, workability, or distensibility” [6]. The nonrenewable resource mineral oil is mostly used in different countries and from which aromatic, naphthenic, and paraffinic oils are produced. Among those, aromatic oil is a very compatible oil with NBR due to its polar properties [7].

However, in 1994 a report by Swedish National Chemicals showed that polycyclic aromatic hydrocarbon (PCA) is the main constituent of aromatic oil, which has been proved harmful for the environment and human health [8].Aromatic process oils (distillate aromatic extracts, DAE)are classified as carcinogenic to humans [9]. Due to treatment, the aromatic content is reduced and these products (treated distillate aromatic extracts, TDAE) are state of the art. TDAE contains 1.3 wt % PCA and according to European legislation, the maximum content of PCA should be 3 wt % [9].

Several researchers have been looking for a sustainable replacement for conventional petroleum oils in the rubber industry. Vegetable oil, such as soybean oil [5], linseed oil [10], castor oil [11], coconut oil [4], and rice bran oil [12], is a renewable and inexpensive oil resource, and research aims to substitute conventional petroleum oils with vegetable oils.

Plasticizers have a specific solubility in rubber and contribute to the Brownian motion of polymer chains; therefore, they reduce the viscosity of the rubber compound [5]. Usually, no chemical reactions take place during the compounding process. The mixing temperature and the kneader-screw-rotation speed influence the mixing properties of the rubber during the mixing process. The temperature of vulcanization of sulfur occurs between 150 °C and 180 °C [13].

The choice of a plasticizer must be matched to the polymer; thus, nonpolar plasticizers are generally used for nonpolar polymers, whereas polar polymers require polar plasticizers [5]. The polar property of NBR thus influences the selection of a plasticizer. The latter is controlled by the ACN content in the NBR backbone chain. Ordinary natural fatty oil is slightly polar so that it becomes ready to react with some active parts to form fatty oil derivatives or to polymerize fatty oils. Moreover, since fatty oils are polar, they are very compatible with NBR (high ACN content). The epoxidized fatty oil contains fewer double bonds compared to conventional fatty oil and contains the active oxirane group. A plasticizer containing OH groups is most compatible with polar polymers, such as NBR [14]. The active sites are available to realize a dipole–dipole interaction with NBR, although the hydrogen bond and Van der Waals bond influences the compatibility. Research was conducted on the plasticization mechanism in the 20th century, and several theories have been postulated on the plasticization mechanism [14,15,16,17].

Khalaf et al. [18] worked with selected vegetable oils as plasticizers for NBR elastomer. They used olive oil and orange oil. The motivation was to find are placement for conventional oil, such as dioctyl phthalate (DOP). Zhu et al. [19] found that the mechanical properties were significantly enhanced by the addition of vegetable oils. Wang et al. [20] used palm oil as the source of renewable plasticizer with ethylene-propylene-diene rubber (EPDM). The investigation revealed that palm oil reduces the Mooney viscosity as well as increases some of the selected mechanical properties. Pechurai et al. [21] worked on castor oil and jatropha oil with styrene-butadiene rubber (SBR). Pechurai showed that the addition of vegetable oils profoundly enhanced the mechanical properties of SBR, and vegetable oils were the correct replacement of petroleum-based oil. Xuan Liu et al. [22] worked on the thermo-oxidative aging of NBR. In their investigation, the authors studied the migration of low-molecular additives as well as chemical changes like post-crosslinking of the materials by using different analytical tests. It was found that during oven aging, post-curing occurred due to high temperatures.

In the present study, epoxidized esters of glycerol formal from soybean and canola oil were used as sustainable, bio-based plasticizers. Two conventional plasticizers, TDAE and Mesamoll^®^, were applied to compare the characteristics. The main components of the vegetable oils are fatty acid esters. The fatty acid contains several double bonds in the fatty acid chains. The epoxidized groups are more active than the double bonds in the fatty acid chains [23]. Thus, it is anticipated that the modified fatty oil is more active than unmodified fatty oil. The active double bonds can play an essential role during the curing of the NBR compounds.

The curing behavior was studied as well as the thermal, mechanical, and dynamic mechanical properties. To investigate the NBR/CB/plasticizer particle dispersion in the NBR compound, an optical analysis was done. Additionally, for selected materials, thermo-oxidative aging was performed.

## 2. Materials and Methods

The investigations were performed on NBR with two different contents of ACN monomer. Arlanexo Deutschland GmbH (Dormagen, Germany) supplied Perbunan^®^ 3445F (34% ACN) and Perbunan® 1846F (18% ACN). The filler was CORAX^®^ N 550 carbon black (CB) from Orion Engineered carbons GmbH (Frankfurt/M., Germany). TDAE was used from Hansen & Rosenthal KG (Hamburg, Germany). Mesamoll^®^ was obtained from Lanxess Deutschland GmbH (Leverkusen, Germany) and bio-oils, and epoxidized ester of glycerol formal from soybean oil (EESO) and epoxidized ester of glycerol formal from canola oil (EECO) from Glaconchemie GmbH (Merseburg, Germany) were used.

### 2.1. Processing of Plasticized Compounds

NBR with 34% ACN and 18% ACN were named NBR-34 and NBR-18 accordingly. The formulation of the materials, which is reported in Table 1, differs mainly in the NBR type, plasticizer types, and contents, whereas the other mixing ingredients were kept almost unchanged.

The compounds were prepared with a two-stage mixing procedure in a lab kneader with a fill factor of 0.7. The starting temperature in the kneader was 50 °C, and a rotation speed of 50 rpm was used. The polymer was introduced first in the chamber, and the CB, plasticizer, stearic acid, ZnO, and 6PPD were added after 1 min. After 5 min, the sulfur and CBS were added. The compounded stock was discharged after 10 min, and as a final step, it was homogenized on a two-roll mill before curing.

Further, cure characteristics at a temperature of 160 °C were determined by using a vulcameter (type GÖTTFERT elastograph, GÖTTFERT Werkstoff-Prüfmaschinen GmbH, Buchen, Germany). According to the results of the vulcameter tests, the time *t_90_* was fixed as the vulcanization time for the necessary plates. For each material (formulation), plates with a size of 120 mm× 120 mm × 2 mm were vulcanized. From these plates, specimens of the desired form for the tests were cut using a metal cutter.

### 2.2. Tests

Hardness (Shore A) of the samples was determined as per DIN ISO 7619 standard [24], using a Zwick hardness tester (type 7206.H04, ZwickRoell, Ulm, Germany). Tensile and tear strength were determined according to ISO 37 [25] and ISO 34-1 [25], respectively, using the universal testing machine (type Zwick Z020, ZwickRoell, Ulm, Germany).

Compression set (CS) values were determined according to DIN ISO 815-1 [25]. Samples with 6.3 mm thickness and 13 mm diameter were compressed to constant strain (~25%) and kept in this condition for 24 h at room temperature. Finally, the CS in % was calculated.
(1)$$\mathrm{CS}\;(\%)=\frac{h_{0}-h_{i}}{h_{o}-h_{s}}\times 100$$

Here, *h_o_* is the thickness of the sample before compression; *h_i_* is the thickness of the sample after recovery, and *h_s_* is the height in mm of the spacer.

The dynamic mechanical analysis (DMA) was carried out on a GABO DMA equipment (type Eplexor 500 N, GABO QUALIMETER Testanlagen GmbH, Ahlden, Germany), according to standard ISO 6721-4 [26]. The storage modulus *E’* and the loss modulus *E”*, as well as the mechanical loss factor tan *δ*, were determined in dependence on the temperature, and the glass transition temperature *T*_g_ was obtained from these data. 

Thermogravimetric analysis of the samples was carried out on a Mettler Toledo instrument with a heating rate of 20 K/min under anitrogen atmosphere up to 600 °C. Then, the atmosphere was changed to oxygen. The following characteristics are determined from the thermograms: the temperature of onset of degradation and the temperature at the peak rate of decomposition, the peak rate of degradation, and the weight of residue remaining at 600 °C. According to ASTM D 2663 [27], the macro filler dispersion was determined. A dispersion index of 100% means that no agglomerate size larger than 6 µm could be found on the cut surface [28]. The samples were prepared by cutting with a sharp metal razor and then investigated with a Leica DM 2700M light microscope.

For the aging test, samples were kept for 500 h at 23°C, 40 °C, and 80 °C. Finally, changes in material behavior were characterized by tensile, hardness, and compression set testing.

## 3. Results

The cure characteristics were examined by using a moving disc rheometer, and the curing curves acquired from NBR-34 compounds with 15 phr of different plasticizers areshown exemplarily in Figure 1. It was clearly evident that, due to the dilution effects [20] of the plasticizers, the maximum torque *M_max_* values decreased when the plasticizers were simply added. *M_max_* is reduced gradually up to 5phr, and a significant reduction has occurred at 15 phr. Minimum torque *M_min_* is comparable, independent of the type of plasticizer, as is shown in Figure 1 for NBR-34. 

Figure 2 gives the torque differences (Δ*M = M_max_* − *M_min_*) of NBR vulcanizate as a function of plasticizer content. The lower *ΔM* indicates one of the possible reasons for the low degree of crosslinking of compounds with a higher loading of plasticizers [4,11,29]. Δ*M* for plasticizer-loaded NBR-18 is comparatively more significant than for NBR-34, as shown in Figure 2. A large number of unsaturated bonds in NBR-18 due to the butadiene monomer could be responsible for the higher degree of crosslinking [30].

The vulcanization time of NBR compounds is shown in Figure 3. The NBR-34 vulcanizates clearly showed a lower curing time compared to the NBR-18 as well as lower difference in torque values (see Figure 2). For NBR-34, the addition of plasticizer changes the vulcanization time, and there is minimal difference between the four plasticizers. TDAE- and Mesamoll^®^-loaded NBR-18 showed increased cure time. However, EECO decreases the vulcanization time of NBR-18. The optimum conversion of epoxidation of bio-based plasticizers is about 90% [31]. As a result, bio-based plasticizers contain 10% fatty acid ester. The acid value of EECO is about 2.32 mg of KOH/mg, whereas the acid value of EESO is about 0.68 KOH/mg [7]. The higher acid value corresponds to the higher free fatty acid. The fatty acid acts as a co-activator during the formulation of NBR compounds [32].

Figure 4 gives the variation of the tensile strength of the NBR vulcanizates with plasticizer loading. For most compounds, the tensile strength is slightly decreased or remains nearly constant with increasing plasticizer content. If 5 phr Mesamoll^®^ and EECO, respectively, are added, the tensile strength increases. This may contribute to more homogenous filler dispersion in the rubber matrix, as reported in [4]. However, our own examinations of the macro-dispersion index gave no correlation with the tensile strength values for increasing amount of plasticizer. The tensile strength of bio-based plasticized NBR-34 is slightly higher compared to NBR-18. One possible reason for this is the higher polarity of NBR-34 due to higher content of ACN compared to NBR-18 [33]. The conventional plasticizers do not contain any OH groups, and therefore, compatibility with a polar polymer-like NBR [14] is restricted, leading to a lower tensile strength.

Generally, a decrease in tensile strength is often combined with an increase in the strain at break [34]. In principle, this is also the case for the materials investigated in this study, as can be seen in Figure 5 showing strain-at-break values of the NBR vulcanizates with different loading of plasticizer. There is a general trend of increasing deformability with rising amount of plasticizer. In detail, some differences can be seen for the different plasticizers. For the NBR-18 compounds, the strain at break remains constant within standard deviation when adding 5 phr of the plasticizers, with the exception of EESO. This may be due to not having optimal filler dispersion or the antiplasticization effect [35].The NBR-34 compounds with bio-based plasticizers have a higher strain at break compared to the compounds with TDAE and Mesamoll^®^. Again, the different polarity of NBR-18 may be the reason for the lower deformability of the materials with the bio-based plasticizers, compared to NBR-34. From the results, a more substantial lubrication effect of the bio-based plasticizers might be derived; in other words, bio-based plasticizers might promote more pronounced polymer chain motion in the vulcanizates [20]. Another reason for the different level of the mechanical properties depending on plasticizer type and amount may be a possible influence on the crosslink density. This was not investigated explicitly, but from the vulcameter curves in Figure 1 it is seen that the maximum torques of the compounds vary, depending on the used plasticizer. This can be a sign of a varying crosslink density. With the currently available data, a chemical reaction between plasticizer and the vulcanization system cannot be excluded, which could lead to a lower crosslink density.

The tear strength values (see Figure 6) show a significant increase in the case of bio-based plasticizers, especially at higher plasticizer loading. In the case of TDAE and Mesamoll^®^, the tear strength remains almost constant at around 8 N/mm. In contrast, the bio-based plasticized NBR compounds showed an increase of up to 9 to 11 N/mm at 15 phr loading range. Tearing of NBR vulcanizate occurs due to the propagation of cracks initiated at the stress concentration point through the wearing of rubber molecules at the NBR–carbon black interfaces [10]. The micro-plasticization of the interfaces using an effective plasticizer can hinder the propagation of cracks [36]. The tear strength is not much influenced by increasing amount of conventional plasticizers. 

Figure 7 shows the variation of hardness with plasticizer concentration. Generally, the hardness is higher without the addition of any plasticizer. With the plasticizers, the materials show continuously decreasing hardness for both types of NBR vulcanizates when increasing the plasticizer content. However, NBR-18 shows lower hardness compared to NBR-34. For both NBR vulcanizates with Mesamoll^®^, highest hardness values were obtained compared to other systems. This is perhaps because of the active participation of Mesamoll^®^ during the vulcanization of NBR. Kukreja [29] stated that the highest hardness might be attributed due to the high participation of crosslinking.

Figure 8 shows the CS values of different oil content of two NBR vulcanizates. Both types of NBR with bio-based plasticizers have a higher CS, which increases with oil content. The use of Mesamoll^®^ and TDAE results in a more or less unchanged CS. A previous study [4] proved that in a rubber vulcanizate with low oil content, the filler dispersion was poor, but at higher oil content, the plasticizing effect and segmental mobility were pronounced, leading to higher CS values [4]. In a previous study [11], compression set values correlated to the hardness values. The highest CS was found for the material with the lowest hardness. Here, the same effect could be found. A low CS value represents a better recovery behavior after load release. However, the bio-based plasticizers show a maximum CS value of ~12%; this is still a low value compared to the threshold, which is ~40% for a gasket [37].

The temperature dependencies of storage modulus *E’* of the plasticizer-loaded NBR vulcanizates are shown in Figure 9. For the values of *E’* in the glassy state all the NBR vulcanizates are similar to each other. In the high-temperature region corresponding to the rubbery state, the NBR vulcanizate without any plasticizer has the highest value of *E’*, followed by NBR/EECO, NBR/Mesamoll^®^, NBR/EESO, and NBR/TDAE vulcanizates (see Table 2). Figure 10 shows the temperature dependences of loss modulus *E”* of various plasticizer-loaded NBR vulcanizates. The *E”* values of all NBR vulcanizates in the glassy state are similar to each other. In the rubbery state, the NBR vulcanizate without any plasticizer shows the highest value. When using plasticizers, *E*’ and *E”* values are decreased. The *E”* values of bio-based plasticizer-loaded NBR vulcanizate are slightly higher (see Table 2). The storage modulus *E*’ decreases with increasing temperature, also after passing the glass transition temperature; this means it is in the entropy-elastic range. This is a typical behavior of carbon-black-filled elastomers and has importance for the use of the material at higher temperatures, e.g., in the case of sealants, where a certain stiffness is necessary.

Figure 11 shows that all the NBR vulcanizates have a single tan *δ* peak. The tan *δ* value converges to a similar value in the low-temperature region. In the high-temperature region, the NBR/bio-based plasticizers have higher values of tan *δ* compared to NBR/TDAE or NBR/Mesamoll^®^. According to previous studies [38,39], the values of tan *δ* and *E’* predicts rubber performance. The maximum of tan δ of the compounds is shifted to lower values in the case of the bio-based plasticizers EESO and EECO as well as the synthetic plasticizer Mesamoll^®^. This is due to the lower *T*_g_ values of these plasticizers. For practical application, a lower *T*_g_ can be of importance, because molecular mobility starts at lower temperatures. This means, the flexibility of the material is given in a larger temperature range. Further, because of the viscoelasticity, a mechanical material loading with higher frequencies can lead to a shift of *T*_g_ to higher temperatures. Therefore, materials with basically lower *T*_g_ are advantageous. The heights of the tan *δ* peak are not strongly different for the different plasticizers. This means, the energy loss in the network is comparable.

The glass transition temperature *T*_g_ was pointed out in Table 2. *T*_g_ is highly influenced by the chemical structure and molecular weight of plasticizers. High molecular-weight plasticizer has high free volume, and high free volume decreases the *T*_g_ [19,20]. Here, the molecular weight of TDAE is around 180 g/mol, and for thebio-based plasticizers it is approximately 400 g/mol.

The thermal stability of plasticizer-loaded NBR vulcanizate was characterized using TGA as a function of various plasticizers and is shown in Figure 12, while the results are summarized in Table 3. NBR vulcanizate without plasticizer shows the best thermal stability. It is noticed that TDAE and bio-based plasticizer-loaded NBR vulcanizates show an almost similar decomposition process and this has improved the first decomposition stage in comparison to Mesamoll^®^; this is because of the ester groups in the bio-based plasticizers [40]. Generally, these plasticizer-loaded NBR vulcanizates show 3 main decomposition steps. Initially, the plasticizer and all other low-molecular components are decomposed up to ca. 350 °C. Then, the polymer is thermally decomposed in the range of 360–500 °C. According to [41], the initial decomposition of CB-reinforced NBR starts from 360 °C. Here, the samples were pyrolyzed to ~32%. After this second decomposition step, the pyrolysis carbon and the filler carbon black remain. The last decomposition step starts at 600 °C under the influence of oxygen and is completed at 650 °C. The decomposition stages are consistent with degradation due to random chain scission of the butadiene and the nature of the ACN parts in NBR vulcanizate [42]. In addition, there are very strong electronegative groups that result in relatively high interaction and high heat resistance in elastomers.

Polarized microscopy was performed as a helpful tool for better understanding the extent of the plasticizers that influenced the dispersion within CB-reinforced NBR vulcanizates. Figure 13 shows the optical pictures of the investigated cross section areas of plasticized NBR vulcanizates. Here, Figure 13a shows a higher number of agglomerates compared to Figure 13b, but the size of agglomerates is smaller. A previous study [43] stated that the plasticizer decreases the shear stress between the CB occluded by the polymer chain. When the polymer is mixed, the polymer molecules have to slip over each other, which is the reason for more agglomerates remaining [15]. Here, the optical observation for Mesamoll^®^- and EESO-loaded NBR vulcanizates indicate a more homogeneous appearance compared to TDAE. TDAE-loaded NBR vulcanizate is depicted in Figure 13c. When the plasticizers were not well dispersed among the matrix, then the number of agglomerates decreased. This phenomenon is called antiplasticization [14]. Figure 13d shows TDAE coagulates within the polymer matrix inhomogeneously near the edge.

Tensile strength after thermo-oxidative aging of plasticizer-loaded NBR vulcanizate is shown in Figure 14. Results show that there is not much influence on tensile strength after thermo-oxidative aging. Shore A hardness consistently increased with increasing aging temperatures (see Figure 15). Figure 16 shows that the compression set decreases with increasing aging time. The reason for changing the mechanical properties after thermo-oxidative aging is perhaps due to the high crosslink formation and oxidated skin, which results from oxygen uptake at the surface of the specimen [44] or post vulcanization [5,45]. Further, the migration of volatile content and plasticizer happens due to long aging time and especially at high temperatures [46]. If the materials are kept at high temperatures for a long time, post-curing could happen during this time [30].

## 4. Conclusions

The use of bio-based plasticizers (EESO, EECO), a conventional plasticizer (TDAE), and a synthetic plasticizer (Mesamoll^®^) in the compounding of NBR vulcanizates were studied. It was shown that bio-based plasticizers can advantageously be used as processing aids instead of conventional oils. Throughout the complete process chain, from preparation of the raw mixture and characterization of the kinetics of the crosslinking reaction to the final properties of the resulting elastomers, positive aspects of the use of bio-based plasticizers were observed, or at least a comparable level of the properties as for the use of mineral oil-based plasticizers. As an example, when using the bio-based plasticizers, there is a certain cure-accelerating effect, and also the crosslinking itself seems to be influenced. The mechanical properties are, generally, a result of the network development during vulcanization, and an improvement or at least a constancy in these properties is one of the main aims of material development. Based on the results shown in this paper, it can be concluded that for the investigated polymers, the use of such sustainable products is possible without noticeable loss in property levels. Future work will focus on other technical elastomers and on understanding of the influence of the plasticizer on the network development.

## Figures and Tables

**Figure 1 materials-13-02095-f001:**
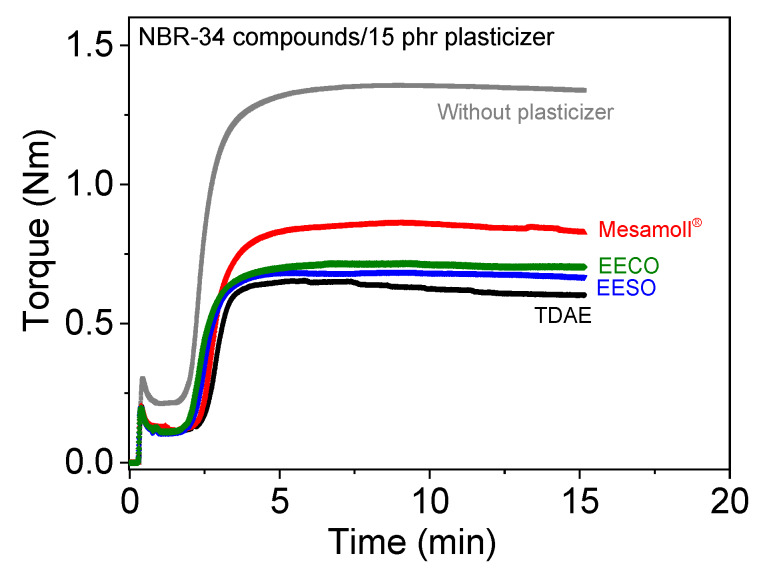
Vulcanization curves of various plasticized NBR compounds.

**Figure 2 materials-13-02095-f002:**
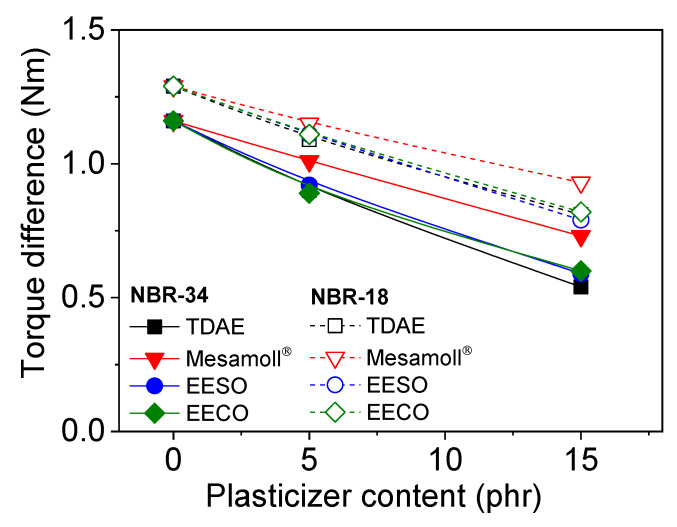
Torque differences during curing of plasticized NBR compounds based on NBR-34 and NBR-18 as a function of plasticizer content.

**Figure 3 materials-13-02095-f003:**
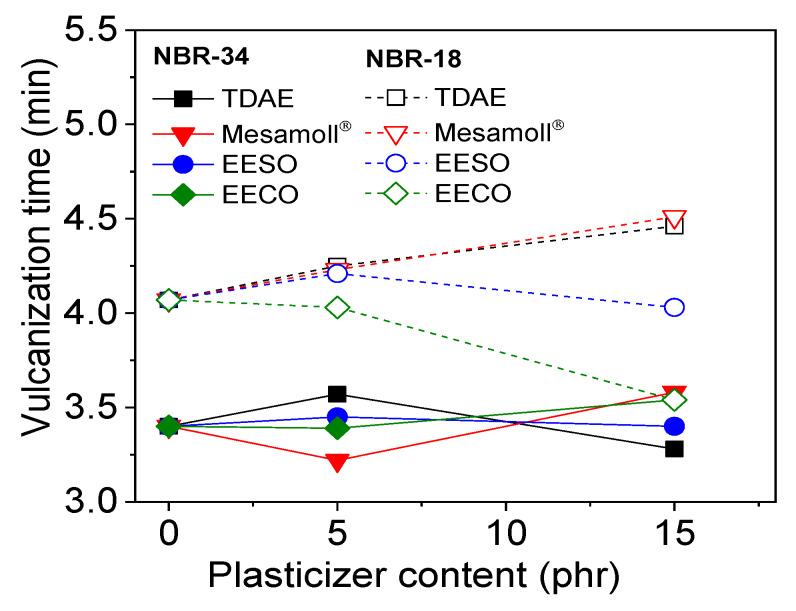
Vulcanization time as a function of plasticizer content for the investigated NBR compounds.

**Figure 4 materials-13-02095-f004:**
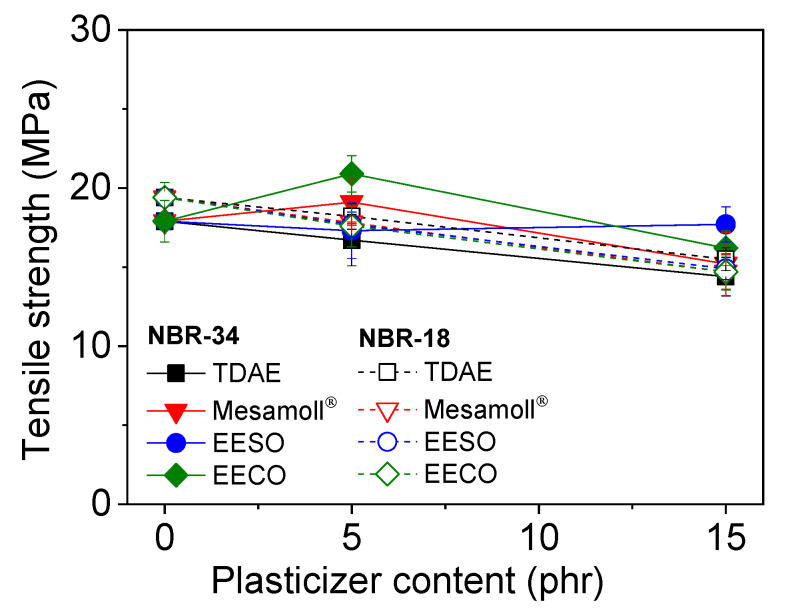
Variation of tensile strength with plasticizer content for NBR vulcanizates.

**Figure 5 materials-13-02095-f005:**
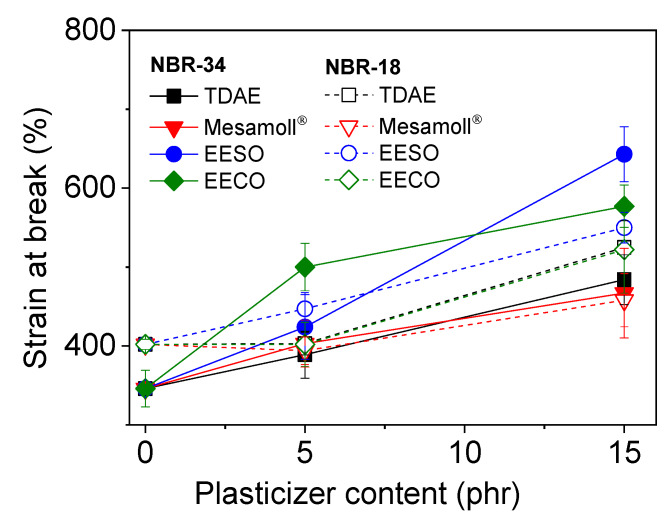
Variation of strain at the break with plasticizer content for NBR vulcanizates.

**Figure 6 materials-13-02095-f006:**
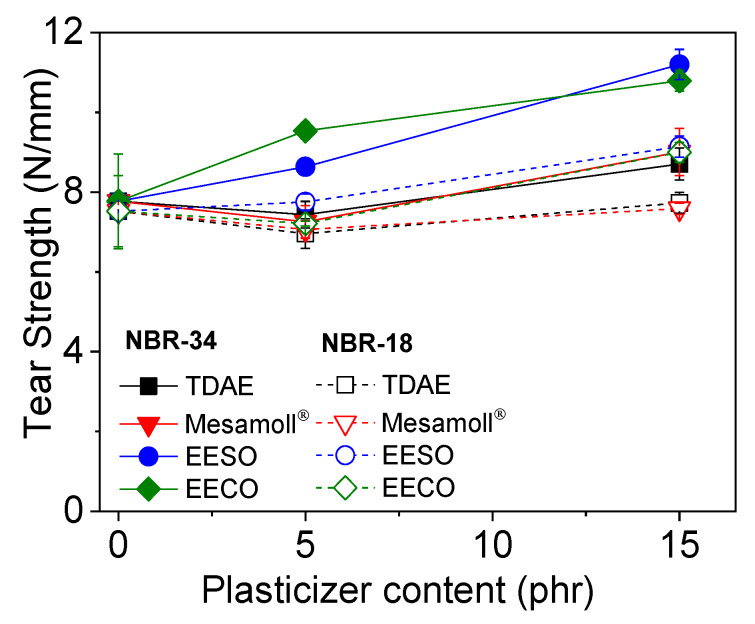
Dependence of tear strength on plasticizer content for NBR vulcanizates.

**Figure 7 materials-13-02095-f007:**
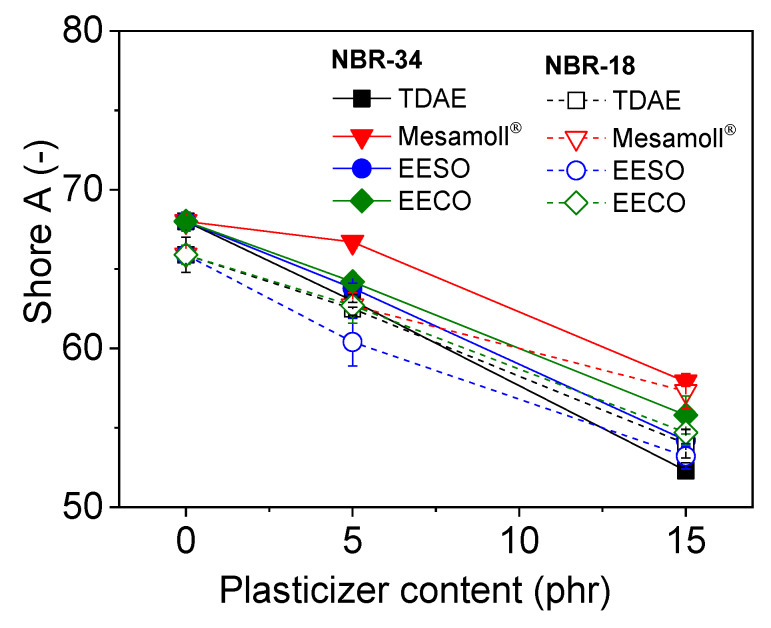
Shore A hardness values as a function of plasticizer content for NBR vulcanizates.

**Figure 8 materials-13-02095-f008:**
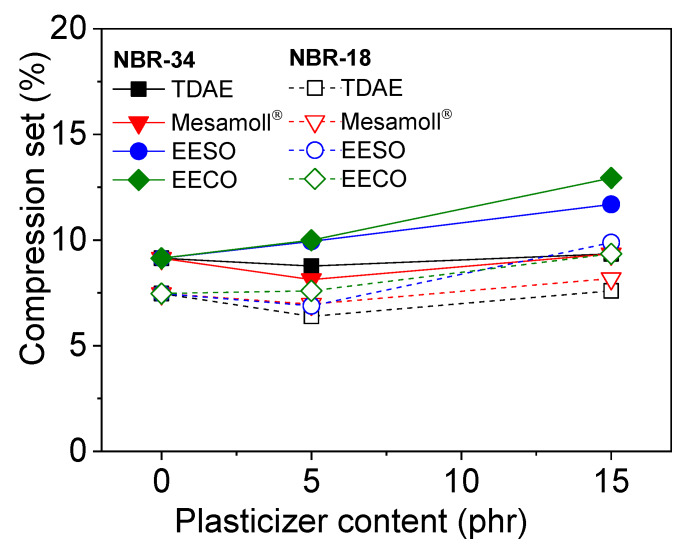
Compression set (CS)values as a function of plasticizer content for NBR vulcanizates.

**Figure 9 materials-13-02095-f009:**
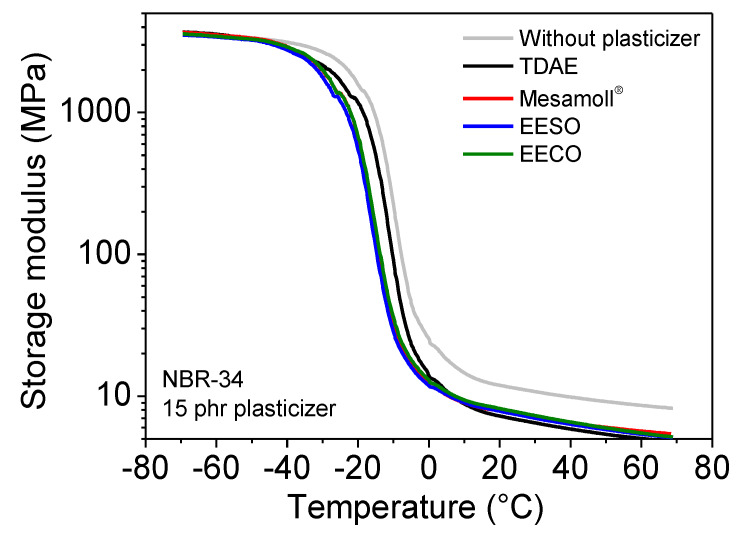
Storage modulus E’ of NBR-34 vulcanizates with 15 phr various plasticizers as a function of temperature.

**Figure 10 materials-13-02095-f010:**
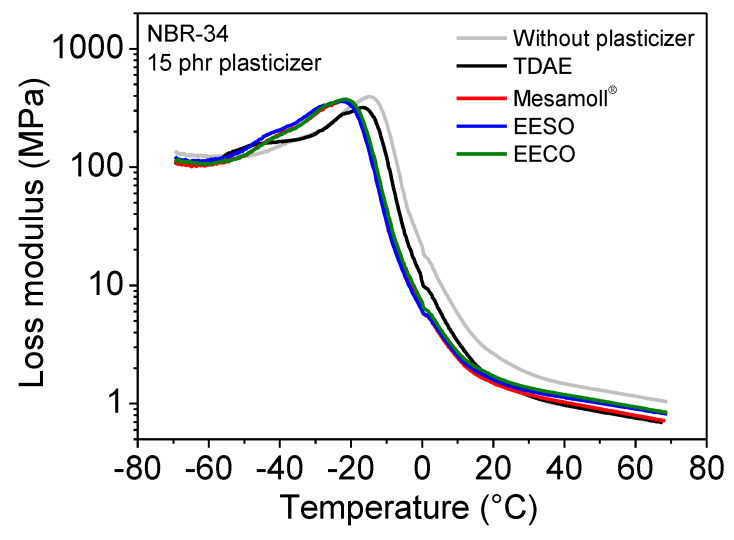
Loss modulus E’’ of NBR-34 vulcanizates with 15 phr various plasticizers as a function of temperature.

**Figure 11 materials-13-02095-f011:**
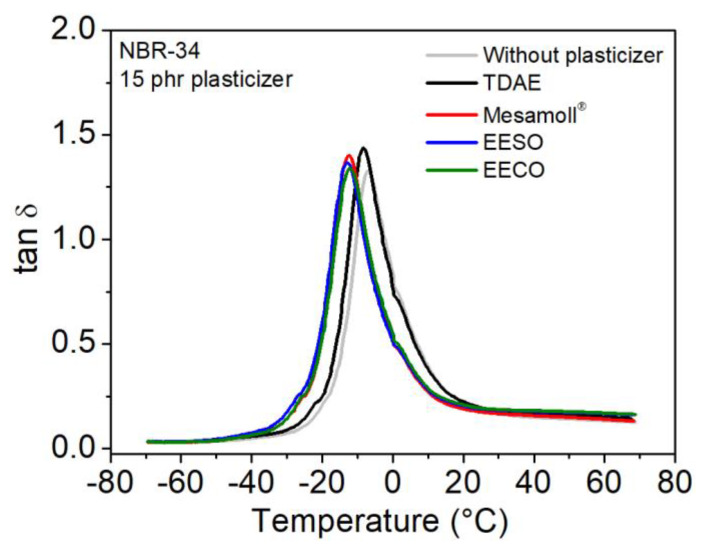
Mechanical loss factor tan *δ* of NBR-34 vulcanizates with 15 phr various plasticizers as a function of temperature.

**Figure 12 materials-13-02095-f012:**
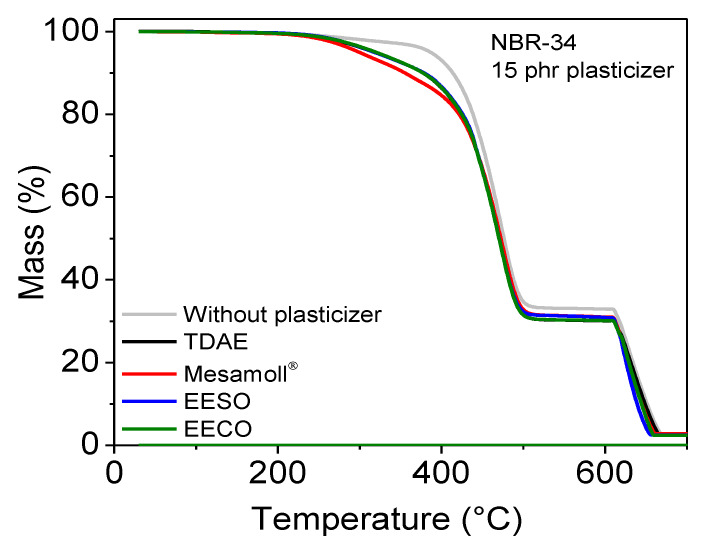
TGA thermograms of NBR-34 vulcanizates with various plasticizers.

**Figure 13 materials-13-02095-f013:**
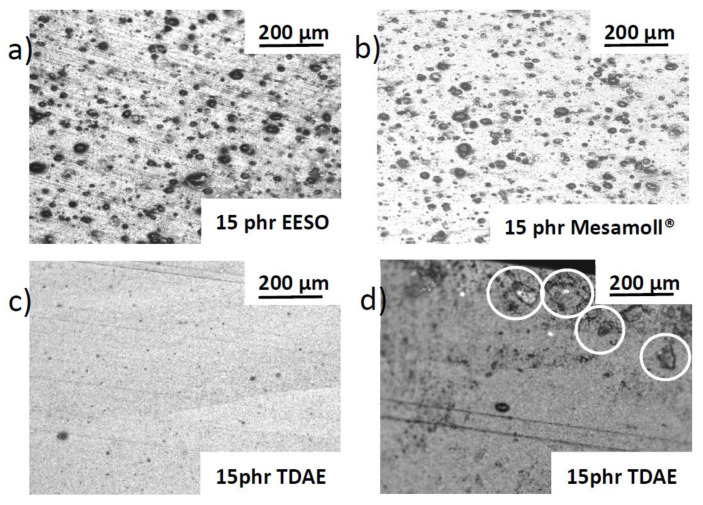
Cross-sectional view of different vulcanized rubber samples; (**a**) NBR-34 vulcanizate with 15 phr EESO, (**b**) NBR-34 vulcanizate with 15 phr Mesamoll^®^, (**c**) NBR-34 vulcanizate with 15 phr TDAE (middle of the samples), (**d**) NBR-34 vulcanizate with 15 phr TDAE (circles mark oil).

**Figure 14 materials-13-02095-f014:**
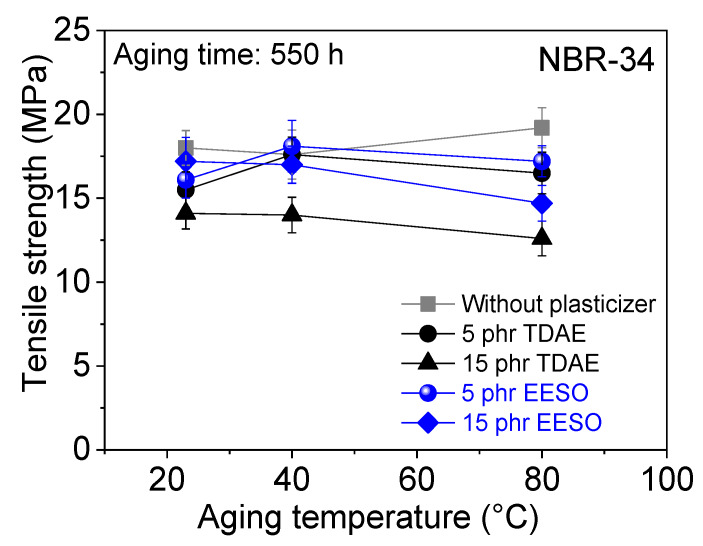
Dependence of tensile strength on aging temperature for NBR-34 vulcanizate with various plasticizers.

**Figure 15 materials-13-02095-f015:**
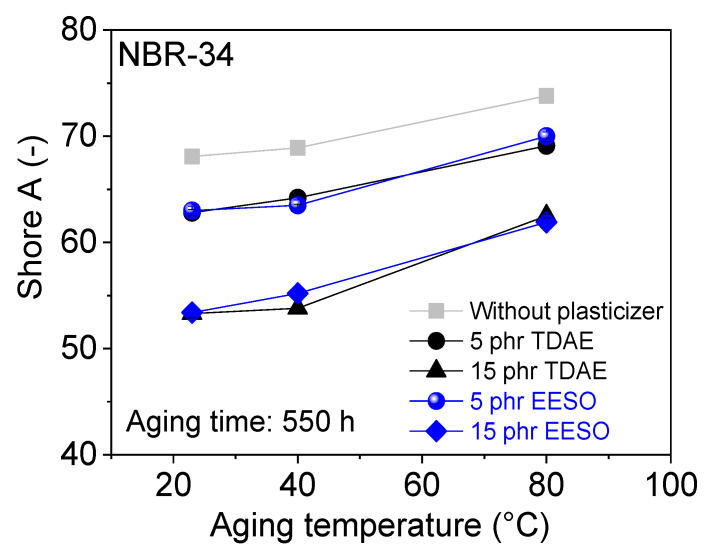
Shore A hardness as a function of aging temperature for NBR-34 vulcanizates with various plasticizers.

**Figure 16 materials-13-02095-f016:**
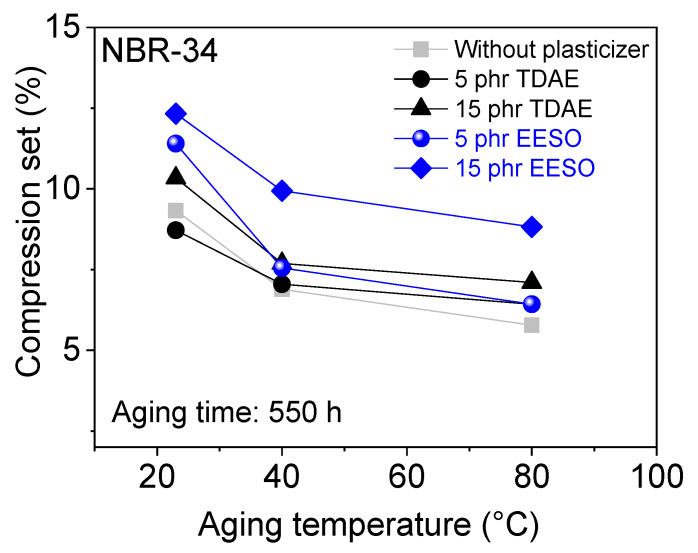
Compression set as a function of aging temperature for NBR-34 vulcanizates with various plasticizers.

**Table 1 materials-13-02095-t001:** Composition and time of addition during mixing.

Name	Comments	Content (phr)	Time of Addition to the Kneader (min)
NBR	34% ACN, 18% ACN	100	0
Carbon black	N 550	40	1
Oil	TDAE, Mesamoll^®^, EESO, EECO	0, 5, 15	1
Stearic acid	Processing aids	1	1
ZnO	Activator	3	1
6PPD^1^	Antioxidant	1.5	1
Sulfur	Crosslinking agent	1.75	5
CBS^2^	Accelerator	1.05	5

^1^ 6PPD, N-(1,3–dimethylbutyl)–N’–phenyl–p–phenylenediamine; ^2^ CBS, N–cyclohexyl–benzothiazole–2–sulfonamide

**Table 2 materials-13-02095-t002:** Dynamic mechanical analysis (DMA) analysis of NBR-34 vulcanizates with various plasticizers.

NBR-34	tan *δ*	*T*_g_(°C)	*E’* (MPa) at 23 °C	*E”* (MPa) at 23 °C
Without plasticizer	1.32	−6.88	11.53	2.31
15 phr TDAE	1.43	−7.34	6.98	1.44
15 phr Mesamoll^®^	1.40	−12.11	7.66	1.37
15 phr EESO	1.36	−14.11	7.55	1.47
15 phr EECO	1.34	−12.56	7.91	1.57

**Table 3 materials-13-02095-t003:** Results of TGA of NBR-34 vulcanizates with various plasticizers.

NBR-34	Decomposition % (250 °C)	Decomposition % (350 °C)	Decomposition % (450 °C)	Decomposition % (550 °C)
Without plasticizer	0.96	2.92	27.55	66.89
15 phr TDAE	1.28	7.4182	33.23	69.74
15 phr Mesamoll^®^	1.73	9.44	33.08	68.66
15 phr EESO	1.17	7.60	33.6	68.72
15 phr EECO	1.28	7.36	33.74	69.66

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
