# Peer review of "Influence of Bio-Based Plasticizers on the Properties of NBR Materials"

_materials, 2020, doi:10.3390/ma13092095_

Round 1

Reviewer 1 Report

The topic of the manuscript is interesting. A wide variety of tests are performed to demonstrate the potential of the biobased plasticizers and are, to my mind, correctly analyzed.

Yet, the quality of English has to be improved, as it sometimes lead to confusing parts of the manuscript.

Apart from that problem of style, I have some few remarks, listed here in order of appearance in the manuscript:

Page 1, line 38: what is exactly meant by « lack of mechanical properties »?

Page 1, first paragraph of Introduction section: please add references to the literature for all the statements

Page 2, sentence from lines 52 to 54 is very confuse, please reformulate

Page 2, lines 54 to 57: this sentence is out of context at this location in the manuscript. It shoud be suppressed or moved elsewhere

Page 2, from line 77: this paragraph appears irrelevant in the present context. Yet, if it is maintained, theories should be presented

Page 2 : sentence lines 92 and 93 is obvious; the nature of properties which are followed upon aging should be specified instead

Page 2, line 94: please recall the exact nature of mentionned modifications

Page 5, lines 188 and 189: there is no proof of that statement (no uniform dispersion of filler at lower  oil loading)

Page 6, line 194: it is surprinsing that an improper dispersion of filler leads to a higher strain at break, please clarify.

Page 7, line 224: it is not clear what is meant by « better elastic properties », please specify.

Author Response

Regarding reviews of Article 764195

Dear Reviewers,
Dear Editor,

we would like to thank you very much for the effort with our paper. Your comments were very valuable for us to improve the quality of the paper. We tried to revise the article in this way that all of your comments were regarded.

In the following, we have answered/commented on the single questions and comments from you.

Best regards,

Mahbubur Rahman

Katja Oßwald
Katrin Reincke
Beate Langer

Response to Reviewer 1 Comments

The topic of the manuscript is interesting. A wide variety of tests are performed to demonstrate the potential of the biobased plasticizers and are, to my mind, correctly analyzed.

Yet, the quality of English has to be improved, as it sometimes leads to confusing parts of the manuscript.

Answer: We have revised the text at various places, and we hope that the quality of the language is better now.

Apart from that problem of style, I have some few remarks, listed here in order of appearance in the manuscript:

Page 1, line 38: what is exactly meant by « lack of mechanical properties »?

Answer: We mean a low level of mechanical properties and so the text was modified.

Page 1, the first paragraph of the Introduction section: please add references to the literature for all the statements

Answer: According to your recommendation we added references (in red color).

Page 2, the sentence from lines 52 to 54 is very confusing, please reformulate

Answer: We reformed the sentences. (Corrected sentenced marked in red color)

Page 2, lines 54 to 57: this sentence is out of context at this location in the manuscript. It should be suppressed or moved elsewhere

Answer: We have deleted those parts.

Page 2, from line 77: this paragraph appears irrelevant in the present context. Yet, if it is maintained, theories should be presented

Answer: We have deleted those parts.

Page 2: sentence lines 92 and 93 are obvious; the nature of properties which are followed upon aging should be specified instead

Answer: The referenced paper was checked again and the information was given regarding the properties.

Page 2, line 94: please recall the exact nature of mentioned modifications

Answer: We have rewritten the nature of modifications. (red color marked for visible the changes)

Page 5, lines 188 and 189: there is no proof of that statement (no uniform dispersion of filler at lower oil loading)

Answer: We checked our experimental results regarding the dispersion index and you are right: that statement is not valid for our result. If we consider the dispersion index as the efficiency of dispersion, then NBR-34/15 phr TDAE  should have a high tensile strength value due to the highest dispersion index. So we overworked this statement.

Page 6, line 194: it is surprising that an improper dispersion of filler leads to a higher strain at break, please clarify.

Answer: We have rewritten these statements.

Page 7, line 224: it is not clear what is meant by « better elastic properties », please specify.

Answer: We have rewritten this text.

Reviewer 2 Report

In Abstact,
line 15, "Here, plasticizers basing on bio or renewable resources, e.g., soybean oil, canola oil, or modifications of these oils, are of interest".
This entence should be re-written because two modified oils were used in the study. and "basing on" -> based on.

There is a contradiction :
In abstact line 25 : "It was observed that the bio-based plasticizers have better aging properties compared to the conventional plasticizers."
In conclusion line 323 : "The thermo-oxidative aging results show that there is no
difference between bio-based plasticizer and TDAE regarding aging-related material changing."

Modified soybean and canola oil are used instead of virgin soybean oil and canola oil in the study. In introduction Line 59, it is mentioned that vegetable oil such as soybean oil, linseed oil, castor oil... have also been studied as in references. Experimental results of the two modified oils should be compared and discussed with unmodified oils along with the conventional oils.

In Figure 5, It needs to be explained why higher ACN content NBR (NBR-34) shows higher strain than NBR-18 for EESO and EECO, but not for traditional ones.

In Figure 6, the author needs to explain or suggest why bio-based plasticizers has higher tear strength. Are they undercured or poorly dispersed as written in the manuscript ? If so, this result could deteriorate the other results in the manuscript.

Author Response

Regarding reviews of Article 764195

Dear Reviewers,
Dear Editor,

we would like to thank you very much for the effort with our paper. Your comments were very valuable for us to improve the quality of the paper. We tried to revise the article in this way that all of your comments were regarded.

In the following, we have answered/commented on the single questions and comments from you.

Best regards,

Md Mahbubur Rahman

Katja Oßwald
Katrin Reincke
Beate Langer

Response to Reviewer 2 Comments

In Abstact,

line 15, "Here, plasticizers basing on bio or renewable resources, e.g., soybean oil, canola oil, or modifications of these oils, are of interest".
This sentence should be re-written because two modified oils were used in the study. and "basing on" -> based on.

Answer: We have rewritten the sentences.

There is a contradiction : 

In abstact line 25 : "It was observed that the bio-based plasticizers have better aging properties compared to the conventional plasticizers." In conclusion line 323 : "The thermo-oxidative aging results show that there is no
difference between bio-based plasticizer and TDAE regarding aging-related material changing."

Answer: These results were a part of my overall work. We had some more results about thermo-oxidative aging, but those were not mechanical or thermal analysis based. When we focused on the fracture mechanics analysis, we found better results when using bio-oils instead of conventional oil. Although we don’t want to show more results here, so we corrected line 25 that results were almost the same.

Modified soybean and canola oil are used instead of virgin soybean oil and canola oil in the study. In the introduction Line 59, it is mentioned that vegetable oil such as soybean oil, linseed oil, castor oil have also been studied as in references. Experimental results of the two modified oils should be compared and discussed with unmodified oils along with the conventional oils.

Answer: We used those statements just for motivation using of bio-oils. We also consider using virgin bio-oils, but results were not noticeable. The researchers who worked with other bio-oils in their materials had different compositions and conditions. So it is hard to compare. Basically, we would like to compare mainly with conventional oils so that we can assess the improving efficiency of specific bio-based plasticizers. Further work, which should be published in the future, concentrates on fracturing properties of such materials with selected, promising bio-based oils.

In Figure 5, It needs to be explained why higher ACN content NBR (NBR-34) shows higher strain than NBR-18 for EESO and EECO, but not for traditional ones. 

Answer: We have rewritten the statements

In Figure 6, the author needs to explain or suggest why bio-based plasticizers have higher tear strength. Are they undercured or poorly dispersed, as written in the manuscript? If so, this result could deteriorate the other results in the manuscript.

Answer: Yes, this may be an aspect, and we added a paragraph, where we mention this possibility. In our last research work, we started with experiments to determine the crosslink density. Unfortunately, it is not useful to do a simple swelling experiment with plasticized materials. Further, because we have a high amount of filler in the material, also NMR measurements are not very simple to perform and analyze.

Reviewer 3 Report

Introduction

The introduction is not well constructed. Due to abundant info, the reader loses the attention and gets further from the main motivation of the manuscript. It is recommended to rewrite the introduction leaner. Unnecessary or the parts degrading the integrity should be removed.

At the beginning of the introduction, it is mentioned that carcinogenic oils should be changed with the vegetable based oils. However, in the last paragraphs of the introduction, it is expressed that modified soybean and canola oil were used instead of virgin soybean oil and canola oil directly. Is it different from the first idea?

Results

In Figure 4, why the curves of tensile strength for EECO for NBR-18 and NBR-34 show different tendencies? What is the reason lays behind for this difference? Figure 4 should be better explained in detail.

It is difficult to understand Figure 5 because of the superposing curves. May be, larger Figure van prevent this. Furthermore, In Figure 5, there is a drop in the strain at break for 5 phr plasticizer content compared to 0 phr ( NBR-18 Mesamoll and EECO). This trend is different from the other curves. Why?

In the results related to DMA, it can be better to give a relationship of  Tg and Mc (crosslink density) ? The explanations for the results of DMA are somehow weak. They should be reinforced and it is better to associate the molecular or mechanistic/thermomechanical mechanisms leading to the differences in between E’, E’’ and tan delta curves. For example, tan delta indicates the damping inside the material but no explanation was found on the manuscript. New references can be added to support your arguments.

Most of the figures on the experimental results have a fatal error. Instead of using curved lines, direct lines should be used. For example, in Figure 14 the line of 5 phr EESO, the curve has an increasing trend after 40 ⁰C then it decreases after 55 ⁰C until the 80 ⁰C. However, you have no data around 55 ⁰C to give such a shape to the line. This should have been corrected by the supervisors of this study.

Conclusion

Conclusion should not be a summary of all your results. It should give a proper message from the results of all your experimental outcomes.

Line14 : Check the structure of the sentence for better understanding

Line 15 : Based on

Line 84: The motivation

Line 85: What is DOP

Line 127: Composition is a better word than recipe

Line 185 : You can use a more formal expression than “more or less”

Line 316 : Less >> Smaller

Author Response

Regarding reviews of Article 764195

Dear Reviewers,
Dear Editor,

we would like to thank you very much for the effort with our paper. Your comments were very valuable for us to improve the quality of the paper. We tried to revise the article in this way that all of your comments were regarded.

In the following, we have answered/commented on the single questions and comments from you.

Best regards,

Md Mahbubur Rahman

Katja Oßwald
Katrin Reincke
Beate Langer

Response to Reviewer 3 Comments

Introduction

The introduction is not well constructed. Due to abundant info, the reader loses the attention and gets further from the main motivation of the manuscript. It is recommended to rewrite the introduction leaner. Unnecessary or the parts degrading the integrity should be removed.

Answer: I tried to rewrite the introduction part; however, I also followed the instructions of other reviewers.

At the beginning of the introduction, it is mentioned that carcinogenic oils should be changed with the vegetable-based oils. However, in the last paragraphs of the introduction, it is expressed that modified soybean and canola oil were used instead of virgin soybean oil and canola oil directly. Is it different from the first idea?

Answer: The general idea came from the replacement of mineral oils with vegetable oils—the pure vegetable oils which I mentioned here as virgin oil. Pure vegetable oil is not compatible with these polymers. Therefore, a chemical modification, e.g., in the form of an epoxidation, is realized to increase the compatibility. During epoxidation, a reaction takes place at the double bonds in the fatty-acid chain oils. So overall, plasticizers from bio-based sources, nevertheless, needs some modifications.

Results

In Figure 4, why the curves of tensile strength for EECO for NBR-18 and NBR-34 show different tendencies? What is the reason lays behind this difference? Figure 4 should be better explained in detail.

Answer: We have rewritten this text.

It is difficult to understand Figure 5 because of the superposing curves. May be, larger Figure van prevents this. Furthermore, In Figure 5, there is a drop in the strain at break for 5 phr plasticizer content compared to 0 phr( NBR-18 Mesamoll and EECO). This trend is different from the other curves. Why?

Answer: We have rewritten this text.

In the results related to DMA, it can be better to give a relationship of  Tg and Mc (crosslink density)? The explanations for the results of DMA are somehow weak. They should be reinforced and it is better to associate the molecular or mechanistic/thermomechanical mechanisms leading to the differences in between E’, E’’ and tan delta curves. For example, tan delta indicates the damping inside the material but no explanation was found on the manuscript. New references can be added to support your arguments.

Thank you for this comment! In our ongoing research work, we will pay attention to this; however, in this paper, we did not see any chance to rewrite the text according to this comment in a concise time. 

Most of the figures on the experimental results have a fatal error. Instead of using curved lines, direct lines should be used. For example, in Figure 14 the line of 5 phr EESO, the curve has an increasing trend after 40 ⁰C then it decreases after 55 ⁰C until the 80 ⁰C. However, you have no data around 55 ⁰C to give such a shape to the line. This should have been corrected by the supervisors of this study.

Answer: Yes, you are right! We say sorry for this, and we changed all the figures and used straight lines between the data points.

Conclusion

Conclusion should not be a summary of all your results. It should give a proper message from the results of all your experimental outcomes.

Line14 : Check the structure of the sentence for better understanding

Answer: We have corrected it.

Line 15 : Based on

Answer: I have corrected it. I got the correction from another reviewer, so I reform this sentence.

Line 84: The motivation

Answer: We have corrected it.

Line 85: What is DOP

Answer: dioctyl phthalate (DOP). We have written it in the article.

Line 127: Composition is a better word than recipe

Answer: We have corrected it.

Line 185: You can use a more formal expression than “more or less”

Answer: We have corrected it.

Line 316 : Less >> Smaller

Answer: We have corrected it.

Reviewer 4 Report

This work investigates the effect of two bio-based plasticizers on the mechanical, viscoelasticity, and thermal properties of acrylonitrile–butadiene rubbers. The results show some improvement in mechanical properties specially in tear strength when using bio-based plasticizers in the formulation. In overall, the manuscript has been well-written, and the results have been clearly presented. Therefore, the reviewer recommends this work to be published in Materials journal after slight modification on English and formatting.

The considered changes are as follows:

  • Line 20: “and so” should be changed to “and therefore,”.
  • Line 131: The two sentences, “Hardness (Shore A) of the samples was determined by using Zwick hardness tester. The used standard was DIN ISO 7619 [23]”, should be merged and rewritten as “Hardness (Shore A) of the samples was determined as per DIN ISO 7619 standard [23], using a Zwick hardness tester”
  • In Figure 1, please include “15 phr plasticizer” on the graph and add a symbol for each line at the end of the line.
  • In Figure 2, the caption should cover NBR-18 as well and symbol names inside the graph, “Bio-oil-1” and “Bio-oil-2”, needs to be modified; same format as other graphs.
  • Line 172: “a lower cure time” should be changed to “a shorter cure time”
  • Line 209: “Generally, hardness is higher without plasticizer addition” to be modified to “without addition of any plasticizer

Author Response

Regarding reviews of Article 764195

Dear Reviewers,
Dear Editor,

we would like to thank you very much for the effort with our paper. Your comments were very valuable for us to improve the quality of the paper. We tried to revise the article in this way that all of your comments were regarded.

In the following, we have answered/commented on the single questions and comments from you.

Best regards,

Md Mahbubur Rahman

Katja Oßwald
Katrin Reincke
Beate Langer

Response to Reviewer 4 Comments

This work investigates the effect of two bio-based plasticizers on the mechanical, viscoelasticity, and thermal properties of acrylonitrile–butadiene rubbers. The results show some improvement in mechanical properties specially in tear strength when using bio-based plasticizers in the formulation. In overall, the manuscript has been well-written, and the results have been clearly presented. Therefore, the reviewer recommends this work to be published in Materials journal after slight modification on English and formatting.

The considered changes are as follows:

Line 20: “and so” should be changed to “and therefore,”.

Answer: We have corrected this.

Line 131: The two sentences, “Hardness (Shore A) of the samples was determined by using Zwick hardness tester. The used standard was DIN ISO 7619 [23]”, should be merged and rewritten as “Hardness (Shore A) of the samples was determined as per DIN ISO 7619 standard [23], using a Zwick hardness tester”

Answer: We have corrected this.

In Figure 1, please include “15 phr plasticizer” on the graph and add a symbol for each line at the end of the line.

Answer: We shifted the text of the legend direct to the curves, so it should be clear which curve belongs to which oil.

In Figure 2, the caption should cover NBR-18 as well and symbol names inside the graph, “Bio-oil-1” and “Bio-oil-2”, needs to be modified; same format as other graphs.

Answer: We have corrected this.

Line 172: “a lower cure time” should be changed to “a shorter cure time”

Answer: We have corrected this.

Line 209: “Generally, hardness is higher without plasticizer addition” to be modified to “without addition of any plasticizer”

Answer: We have corrected this.

Round 2

Reviewer 2 Report

The author's reply is appropriate. So, I think it is now publishable in the journal. 

Author Response

Dear Reviewer,

We would like to thank you very much for the effort with our paper. We are accepting your cordial feedback.

Best regards,

Md Mahbubur Rahman

Katja Oßwald
Katrin Reincke
Beate Langer

Reviewer 3 Report

You should still improve your discussion on DMA part. You need to also add your comments and discuss the results to contribute the scientific field.

Conclusion was not modified according to my previous recommendations.

Line 183, own >> our ?

Author Response

Dear Reviewer,

we would like to thank you very much for the effort with our paper. Your comments were very valuable for us to improve the quality of the paper.

Best regards,

Md Mahbubur Rahman

Katja Oßwald
Katrin Reincke
Beate Langer

Response to Reviewer 3 Comments_round 2

You should still improve your discussion on the DMA part. You need to also add your comments and discuss the results to contribute to the scientific field.

Answer: We have tried to improve the DMA part according to your recommendation.

The conclusion was not modified according to my previous recommendations.

Answer: We have changed some points in the conclusion.

Round 3

Reviewer 3 Report

Former recommendations are considered.